# A Finite Element Analysis of Tunnel Lining Demolition by Blasting for Subway Tunnel Expansion

**Jie Zhou [1], Pengyu Shu [2], Bin Zhang [1,3,4,*], Baowang Deng [2] and Yi Wu [3]**

[1]  State Key Laboratory of Mountainous Bridge and Tunnel Engineering, Chongqing 400074, China
[2]  China Construction Bridge Co., Ltd., Chongqing 402218, China
[3]  School of Engineering and Architecture, Chongqing University of Science and Technology, Chongqing 401331, China
[4]  T.Y. Lin International Engineering Consulting (China) Co., Ltd., Chongqing 401121, China
*  Correspondence: 2020007@cqust.edu.cn; Tel.: +86-023-65023977

**Abstract:** In this paper, a practical project of subway tunnel lining demolition via blasting for the construction of a subway station under the action of the blasting load and the weight of collapsed rock mass was proposed. The tunnel overbreak and underbreak quality, the failure mechanism of the tunnel lining structure, the particle peak velocity (PPV), and the stress evolution law of the surrounding rock caused by tunnel blasting were researched using LS-DYNA. Firstly, the results show that the blasting parameters presented in this paper can maintain the cross-section of a smooth outline of tunnel excavation and the overbreak or underbreak quality in control. Secondly, the tensile stress in the existing tunnel lining caused by blasting exceeded the concrete tensile strength, and the radius of the burst fracture was 0.86 m, which is greater than the thickness of the tunnel lining (0.7 m). Thirdly, the blasting stress in the surrounding rock peaked within $0.1 \times 10^{-3}$ s after the blasting, and failure of the surrounding rock occurred. Moreover, the relationship between the PPV and the distance from the blasting center shows that the blasting parameters used in this paper can effectively control the PPV. Therefore, this study reveals that the expansion of existing tunnels into subway stations using this method can improve the efficiency of construction.

**Keywords:** subway tunnel; expansion excavation; tunnel lining demolition; blasting load; failure mechanism; particle peak velocity (PPV)

## 1. Introduction

The rapid development of cities will not only lead to an increase in pedestrian density, but it may also change the planning function of certain areas along subway lines. Therefore, subway stations need to be added after the subway tunnels have been built if the subway station configuration is not reasonable. Since the size of the cross-section of existing subway tunnels is smaller than that of the platform halls of the subway stations, it is necessary to expand the subway tunnels to form new subway stations. The construction process involves two problems: one is determining how to quickly demolish the existing subway tunnel lining, and the other is the safety of the expansion excavation of the existing subway tunnel. These two difficulties put forward extremely high requirements for construction technology [1–3]. At present, the most common demolition methods of tunnel lining are the use of machines or blasting. However, the efficiency of the mechanical demolition method is very slow, the risk of sudden lining collapse is huge, and blasting demolition needs a large number of explosives; thus, the shock waves produced by blasting cause great damage to the tunnel lining structure of the adjacent non-expanded tunnel and increase the risk of tunnel construction [4–6]. Consequently, the common methods of lining demolition are not the best options.

However, drilling and blasting are the most popular methods of tunnel expansion construction, and there are few cases of subway tunnel in situ expansion excavation to

a subway station by drilling and blasting. Considering newly built tunnel excavation and conventional tunnel in situ expansion engineering cases, some research achievements have been published regarding the influence of blasting shock waves on the surrounding rock and the tunnel lining structures of adjacent non-expanded tunnels. The mechanical responses of the surrounding rock and the tunnel structure have been studied during tunnel blasting excavation [7–16]. For conventional tunnel in situ expansion engineering, Liu et al. [17] determined the blasting vibration velocity law with multiple free surfaces by conducting a numerical analysis. Zhao et al. [18] analyzed the vibration velocity of the key parts of a tunnel structure by comparing the results from a numerical simulation and a field measurement of tunnel expansion using blasting. Zhang et al. [19–22] proposed a basis for blasting safety evaluation based on the particle vibration acceleration index according to the impact of tunnel expansion using blasting on the existing tunnel lining. The aforementioned research achievements of the mechanical response of the surrounding rock and lining structure in the drilling and blasting construction process can be used as references in the study of blasting mechanical characteristics analysis if the subway station was expanded from an existing subway tunnel using in situ blasting.

In this paper, based on the project background of the in situ blasting expansion of a subway tunnel to a subway station, a method of rapid removal of the tunnel lining is put forward, which aims to crush the existing pre-cut tunnel lining under the action of blasting shock waves and the weight of collapsed rock mass during the tunnel in situ expansion excavation process. The overbreak and underbreak of the tunnel excavation outline, the blasting response during the tunnel expansion construction, the stress evolution of the surrounding rock caused by the blasting shock waves, and the vibration velocity of the surrounding rock were analyzed according to a numerical simulation of a subway tunnel expanded in situ to a subway station by drilling and blasting. The findings can provide theoretical support to determine the blasting parameters of in situ expansion excavation during the subway station construction process.

## 2. Project Profile

After the completion of the subway tunnel construction in Chongqing, a subway station needs to be added due to the functional positioning of a particular area along the metro line being changed into a business district. Therefore, there arises a special engineering case of a subway station formed by the in situ expansion excavation of an existing subway tunnel. The newly added subway station is the side platform type with a separate hall and platform. The station is located in a straight subway line, and the length is 176 m, as shown in Figure 1. The excavation size of the station is 13.42 m in width and 11.16 m in height; the expansion scheme of the existing subway tunnel section to the platform section is shown in Figure 2. The subway station is in IV-class surrounding rock, which is a tectonic denudation landform, and the geological conditions are unfavorable for tunnel construction. It is a shallow-buried underground structure with a depth of 47.2 m, and the existing adjacent structures, such as the slope retaining wall and non-expanded subway tunnel lining, require a high-level blasting parameter design to obtain good excavation quality and a low blasting vibration velocity. Moreover, the short time limit of the project requires that the existing tunnel lining demolition be performed quickly to ensure that the subway station construction is completed on time. Therefore, the rapid demolition lining method is proposed by using the surrounding rock blasting load and the weight of the collapsed surrounding rock to crush the pre-cut lining during the in situ expansion.

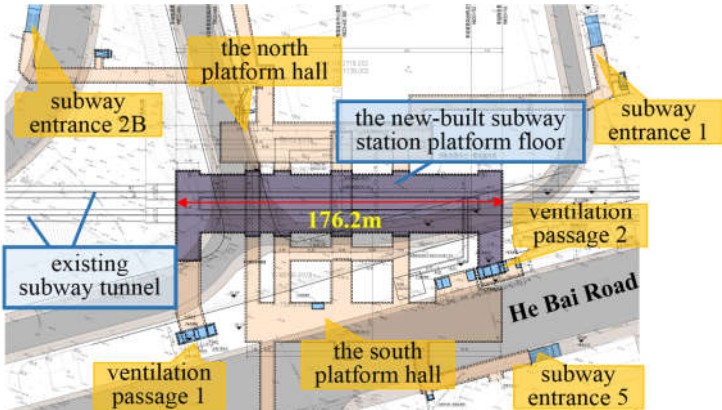

**Figure 1.** The plan of subway tunnel in situ expansion excavation to a subway station.

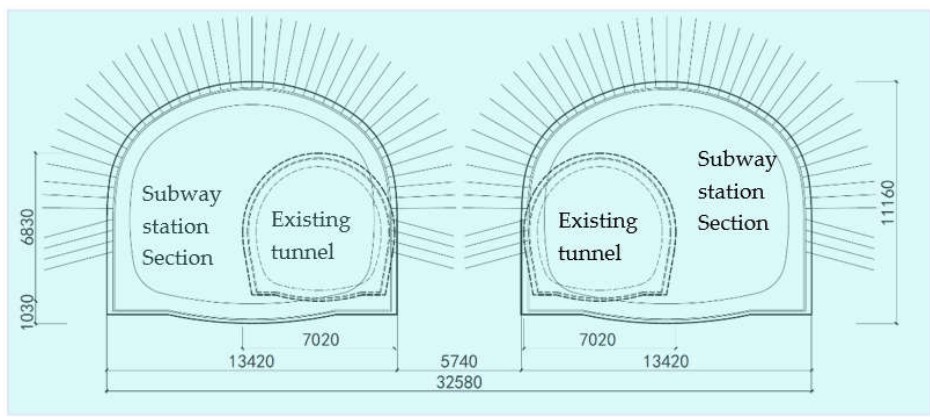

**Figure 2.** A schematic diagram of in situ expansion excavation method for subway station (unit: mm).

## 3. Parameter Design of Blasting Expansion

The thickness of the existing subway tunnel lining structure is 70 cm, which needs to be cut in longitudinal and circular directions by a machine, and then the existing subway tunnel's cross-section needs to be expanded using the drilling and blasting method with optimized blasting parameters; thus, the existing tunnel lining will be crushed by blasting shock waves and the collapsed surrounding rock mass. In order to achieve the purposes of quickly demolishing the tunnel lining and reducing the expansion blasting load influence on the adjacent non-expanded subway tunnel structure, the center diaphragm (CD) excavation method is adopted for the construction of a subway station that is in situ expanded from an existing subway tunnel. The construction process is as follows:

(1) Cut the existing subway tunnel lining, and the average spacing of the longitudinal and circular cutting joints is 3 m and 1.5 m, respectively.

(2) Expand the excavation of the left drift heading of the station section using the drilling and blasting method, and support the excavated left drift heading using an anchor and shotcrete.

(3) Expand the excavation of the right drift heading of the station section using the drilling and blasting method, and support the excavated right drift heading using an anchor and shotcrete.

(4) Excavate and support the inverted arch.

(5) Build the tunnel lining structure of the platform section after the expansion excavation process.

The subway tunnel station is in IV-class surrounding rock mass of relatively broken sandy mudstone, and the value of compressive strength is 24.7 MPa. The blasting footage of the existing subway tunnel in the expansion construction is 1.5 m, which can ensure the safety of subway station construction. Moreover, the distance between the adjacent

non-expanded subway tunnel structure and the expanded blasting surface is about 80 m. Some research results suggest that the blasting vibration velocity must be controlled within 5.0 cm/s during the blasting process [3,4,7]. The permitted maximum explosive charge of each excavation cycle is 25.6 kg in the CD excavation method of the tunnel construction process, which is calculated by the formula of Sodev's maximum vibration velocity shown in formula 1 [4,7,15].

$$V = K(\frac{Q^{1/3}}{R})^{\alpha} \tag{1}$$

where $V$ is the maximum vibration velocity (unit: cm/s), $Q$ is the explosive charge quality in each blasting cycling (unit: kg), $R$ is the distance between the blasting face and the control particle (unit: m), $K$ is the site coefficient, and $\alpha$ is the attenuation coefficient.

The explosive charge in the controlled blasting construction is listed in Table 1, which is determined using Equation (1). In order to achieve a smooth blasting excavation outline and a small vibration velocity, it is necessary to design the blasting hole spacing. Based on a numerical simulation calculation, the optimized blasting parameters are determined as follows: the spacing between peripheral boreholes is 50 cm, the spacing between auxiliary holes is 80 cm, and the spacing between the bottom holes is 70 cm. The optimized borehole layout is shown in Figure 3.

**Table 1.** The permitted maximum explosive charge per detonating period.

| Expansion Method | Maximum Control PPV $V$/cm·s$^{-1}$ | $K$ | $\alpha$ | Distance between Blasting Center and Control Particle $R$/m | Charge Quantity of Each Cycling Blasting $Q$/kg |
|---|---|---|---|---|---|
| The CD excavation method | 5.0 | 250 | 1.5 | 40 | 25.6 |

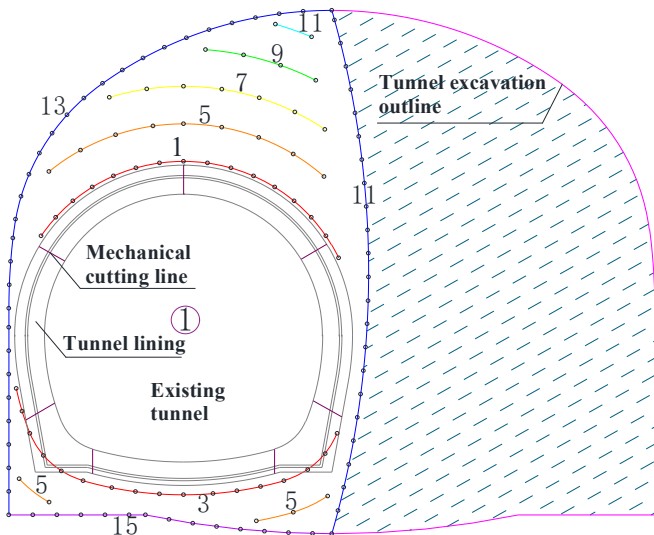

**Figure 3.** Blast holes of subway station expansion excavation.

The diameter of the borehole is 42 mm, and using the No.2 rock ammonium nitrate explosive, the density is 1.05~1.25 g/cm$^3$, the explosion intensity is 295 mL, the explosive intensity is 12 mm, and the detonation velocity is 5000 m/s in the blasting process. Moreover, a millisecond detonator is used to initiate the explosive. The explosive charges per 1.5 m of excavation footage are listed in Table 2.

**Table 2.** The explosive charge for the left drift heading excavated by the CD excavation method.

| Hole Category | The Segments of Detonator(s) | Number of Holes | Charge Collection (kg/m) | Charge Quantity Per Hole (kg) | Dosage Per Dose (kg) |
|---|---|---|---|---|---|
| Relief holes | 1 | 16 | 0.67 | 1 | 16.0 |
| | 3 | 17 | 0.67 | 1 | 17.0 |
| | 5 | 14 | 0.33 | 0.5 | 7.0 |
| | 7 | 7 | 0.33 | 0.5 | 3.5 |
| | 9 | 4 | 0.33 | 0.5 | 2.0 |
| | 11 | 24 | 0.33 | 0.5 | 12.0 |
| Trim holes | 13 | 30 | 0.23 | 0.35 | 10.5 |
| Bottom holes | 15 | 15 | 0.33 | 0.5 | 7.5 |
| Total | - | 127 | - | - | 75.5 |

## 4. Blasting Mechanical Characteristics

### 4.1. Numerical Calculation Model and Parameters

A three-dimensional model is established for simulation using LS-DYNA. In order to reduce the boundary effect and ensure the accuracy of the calculation, the dimensions of the numerical model are as follows: the boundaries in the X, Y, and Z directions are 120 m, 70 m, and 1.5 m, respectively, as shown in Figure 4.

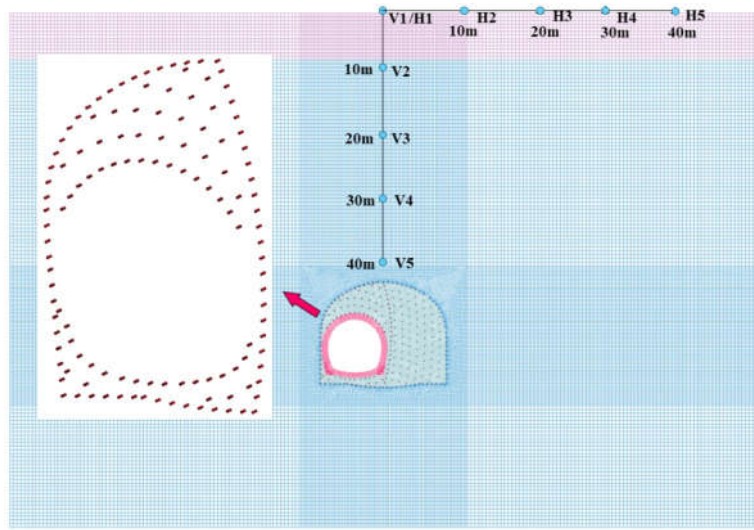

**Figure 4.** Numerical model of subway station using the CD excavation method.

The lateral boundary of the numerical model is subject to normal constraints in the horizontal direction, the bottom is subject to fixed constraints in the vertical direction, and the non-reflection boundary conditions of all boundaries are set to simulate the semi-infinite boundary of the blasting [23]. In order to approach the actual cutting seam of the existing pre-cut subway tunnel lining in the simulation, the element strength reduction method is adopted to set the cutting seam element material to 0.1 times that of the tunnel lining material [24]. For a more convenient dimensional analysis and calculation, the unit is cm-g-ms in the modeling process [19].

The in situ expansion excavation of a subway tunnel to a subway station is carried out using the CD excavation method. The existing subway tunnel lining structure of the left drift heading is demolished after the left drift heading excavation step (i.e., the first step of the expansion excavation). Therefore, the excavation free face during the right drift heading excavation is more than that during the left drift heading excavation step. Considering the excavation free face can attenuate the propagation of blasting waves; the most dangerous blasting step occurs in the left drift heading excavation step. This paper only calculates and analyzes the left drift heading excavation of the CD excavation method [1–4].

Plastic mechanical models are adopted for the rock mass and concrete, and the C30 concrete material is defined as MAT_PLASTIC_KINEMATIC in the numerical model. The material parameters of the calculation model are shown in Table 3, and the physical parameters of the No.2 rock ammonium nitrate explosive are shown in Table 4. The relationship between the unit pressure and volume (P-V) of the detonation products of high-energy explosives is described by Equation (2), which can simulate the explosive explode state accurately.

$$P = A\left(1 - \frac{\omega}{R_1 V}\right)e^{R_1 V} + B\left(1 - \frac{\omega}{R_2 V}\right)e^{-R_2 V} + \frac{\omega E}{V} \tag{2}$$

where $V$ is the relative volume; $E$ is the internal energy constant; and $A$, $B$, $R_1$, $R_2$, and $\omega$ are state equation coefficients.

**Table 3.** Numerical parameters of surrounding rock and tunnel support structure.

| Material Number | Material Name | Elastic Modulus $E$ (MPa) | Poisson's Ratio $\mu$ | Bulk Density $\gamma$ (kN·m$^{-3}$) | Cohesion $c$ (MPa) | Internal Friction Angle $\varphi(°)$ |
|---|---|---|---|---|---|---|
| 1 | Mudstone | $1.2 \times 10^3$ | 0.27 | 22 | 0.6 | 22 |
| 2 | Tamping plug | $0.5 \times 10^3$ | 0.3 | 20 | 0.25 | — |
| 3 | Preliminary lining concrete | $2.1 \times 10^4$ | 0.18 | 24 | — | — |
| 4 | Inverted arch concrete | $2.1 \times 10^4$ | 0.18 | 24 | — | — |
| 5 | Tunnel lining concrete | $3.15 \times 10^4$ | 0.18 | 24 | — | — |

**Table 4.** Parameters of high-energy explosive materials.

| $\rho$/(g·cm$^{-3}$) | $A$ (GPa) | $B$ (GPa) | $R_1$ | $R_2$ | $\omega$ | $E$ (GJ·m$^{-3}$) |
|---|---|---|---|---|---|---|
| 1.05 | 210 | 0.2 | 4.2 | 0.95 | 0.15 | 4.13 |

*4.2. Results Analysis*

4.2.1. Blasting Quality Analysis

The numerical calculation adopts the element failure method; the element automatically exits the calculation when its stress reaches the material limit stress. That is, tunnel lining failure occurs when the compressive strength is more than 20.1 MPa or when the tensile strength is more than 2.01 MPa. According to the construction steps of the subway station newly built following the method of in situ expansion excavation of the subway tunnel, the demolition effect of the existing tunnel lining for the left drift heading at different times during the blasting process is shown in Figure 5.

The calculation results show that damage began to occur to the pre-cut tunnel lining structure at $0.4 \times 10^{-4}$ s after the dynamite blast, and it was completely crushed at $0.8 \times 10^{-4}$ s after the dynamite blast. The existing tunnel lining was demolished by a blasting load within a short time. A comparison of the actual excavation outline with the design excavation contour line after the tunnel blasting using the CD excavation method, which can determine the amount of tunnel overbreak and underbreak of the excavation and evaluate the blasting effect, is shown in Figure 6.

Using the blasting parameters in Table 2 and the CD excavation method, the subway tunnel is expanded into a subway station by blasting. It can be determined from Figure 6 that the maximum tunnel overbreak in the vault area, side wall, and inverted arch area are 15 cm, 7 cm, and 5 cm, respectively. The numerical simulation results show that the maximum tunnel overbreak value in the vault area is 15 cm, which is lower than the requirements for the excavation quality of subway tunnel cross-sections defined in the Quality Acceptance Standard for Subway Construction (GB/T 50299-2018) [25]. Therefore, the subway tunnel lining will collapse under the action of blasting impact stress and the gravity of the surrounding rock, which makes it possible to be removed quickly.

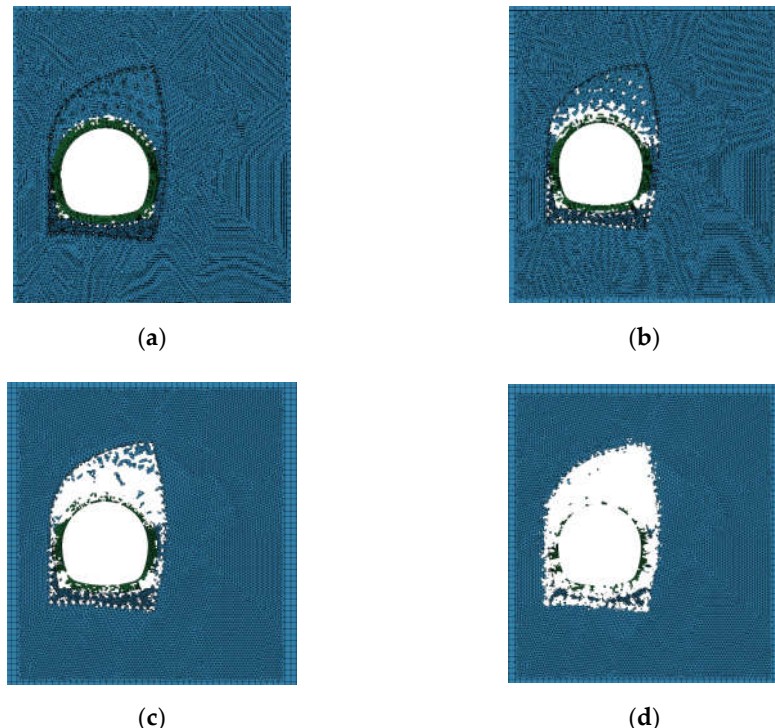

(a) (b)

(c) (d)

**Figure 5.** Sate of tunnel lining and rock in blasting process: (**a**) $T = 0.4 \times 10^{-4}$ s; (**b**) $T = 0.16 \times 10^{-3}$ s; (**c**) $T = 0.4 \times 10^{-3}$ s; (**d**) $T = 0.8 \times 10^{-3}$ s.

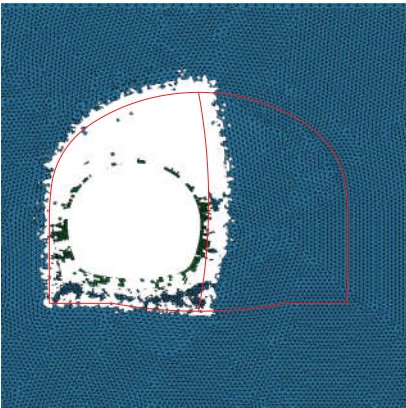

**Figure 6.** Expanded tunnel blasting effect caused by the CD excavation method.

### 4.2.2. Failure Mechanism of the Vault of Tunnel Lining

Demolishing the vault of the existing tunnel lining is the most critical step in the whole subway station construction process; subsequent construction is more convenient after the demolition of the tunnel lining structure. The stress of the vault of the tunnel lining at different times during the blasting process is shown in Figure 7.

It can be seen in Figure 5a that the vault area of the tunnel lining is damaged at the time of $0.4 \times 10^{-4}$ s. It can be seen in Figure 7a that the maximum principal stress $S_1$ (tensile stress) and $S_3$ (compression stress) of the damaged area are 10.6 MPa and 15.0 MPa, respectively. According to the principal stress nephogram analysis, the maximum principal stress $S_1$ and $S_3$ of the tunnel lining structure first increase and then decrease with the blasting time. During the blasting process, the maximum principal stress $S_1$ is 35.6 MPa at the time of $0.16 \times 10^{-3}$ s, and the maximum principal stress $S_3$ is 45.3 MPa at the time of $0.4 \times 10^{-3}$ s. Combined with the damaging effect of the tunnel lining in the vault area shown in Figure 5, it is convincing that tensile stress plays a leading role in damaging the existing subway tunnel lining structure.

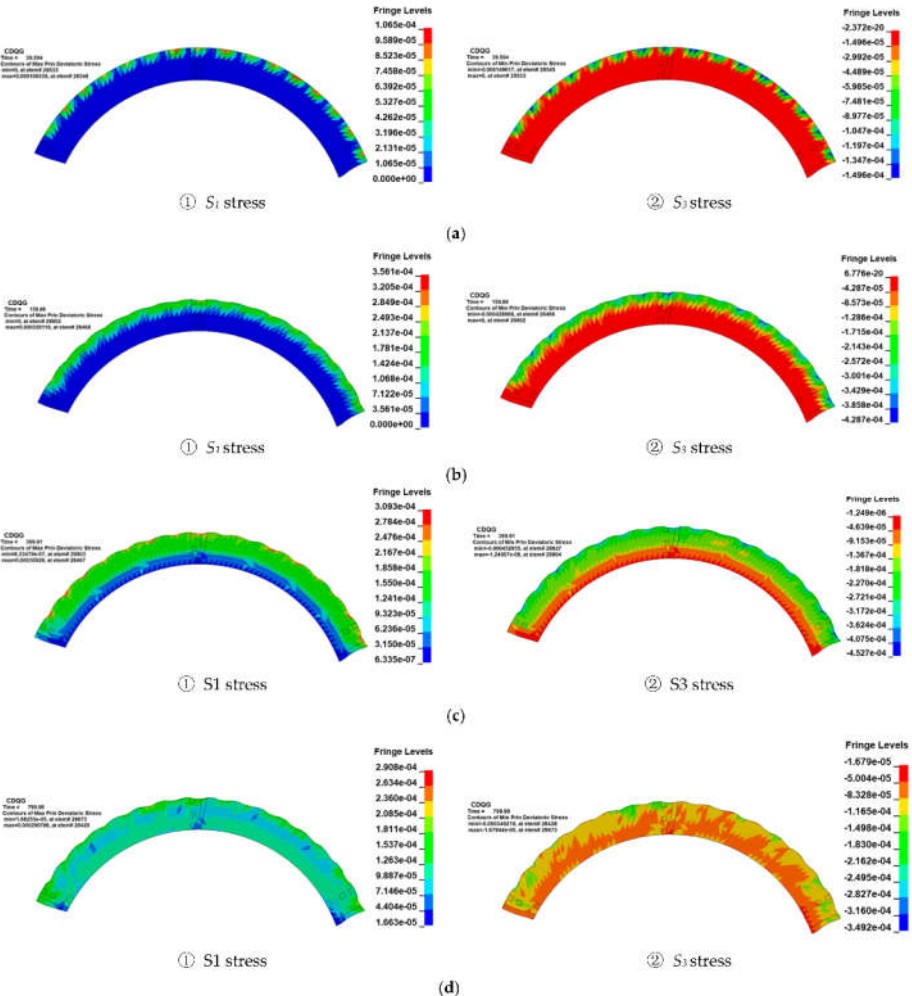

**Figure 7.** Principal stress nephogram at different times after blasting (unit:$10^{11}$ Pa): (**a**) principal stress nephogram at time of $0.4 \times 10^{-4}$ s; (**b**) principal stress nephogram at time of $0.16 \times 10^{-3}$ s; (**c**) principal stress nephogram at time of $0.4 \times 10^{-3}$ s; (**d**) principal stress nephogram at time of $0.8 \times 10^{-3}$ s.

(1)  Stress analysis of neutral axis of tunnel lining

Through the numerical simulation results, a stress evaluation of the outer side, neutral axis, and inner side along the thickness of the lining can be carried out, as shown in Figure 8. The maximum stress of the aforementioned three measuring points on three sections (i.e., the left and right arch footing sections and the vault sections) are listed in Table 5.

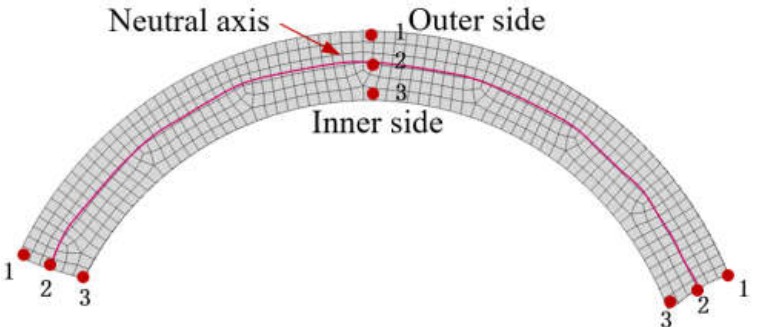

**Figure 8.** The measuring points of tunnel lining.

**Table 5.** The maximum stress of typical section of tunnel lining (unit: MPa).

| Measuring Points | | Vault | | Left Arch Footing | | Right Arch Footing | |
|---|---|---|---|---|---|---|---|
| | | $S_1$ | $S_3$ | $S_1$ | $S_3$ | $S_1$ | $S_3$ |
| Outer side | 1 | 28.6 | 43.7 | 19.3 | 20.3 | 24.8 | 27.8 |
| Neutral axis | 2 | 17.4 | 17.0 | 14.5 | 17.1 | 18.6 | 16.6 |
| Inner side | 3 | 13.8 | 12.8 | 13.6 | 10.5 | 13.4 | 8.91 |

It can be seen in Table 5 that the maximum tensile stresses of all the nine points are greater than the tensile strength $\sigma_t$ (2.01 MPa), and the maximum compassion stresses of three points are greater than the compressive strength $\sigma_f$ (20.1 MPa). Thus, this indicates that the tunnel lining is damaged because of the combined action of the tensile stress and compressive stress based on the maximum tension theory and the maximum strain energy density theory, but tensile failure is the essential factor.

(2) Radius of blasting fracture zone

The circumferential tensile stress, tangential tensile stress, and gas quasi-static pressure produced by the blasting stress wave work together to create a fracture zone. The calculation equation of the radius of the blasting fracture zone is as follows:

$$R_P = r_b(\psi p/\sigma_t)^{1/\alpha} \tag{3}$$

where $R_p$ is the radius of the blasting fracture zone (unit: m); $r_b$ is the radius of the blast hole (unit: mm); $\sigma_t$ is the tensile strength of the rock (unit: Pa); and $\psi$ is the lateral stress ratio, it can be obtained by Equation (4):

$$\psi = \mu/(1-\mu) \tag{4}$$

In Equation (3), $p$ is the peak value of the initial pressure on the wall of the blast hole (unit: Pa), and it can be obtained using Equation (5):

$$p = \rho_s D^2 (r_c/r_b)^6/8 \tag{5}$$

where $\rho_s$ is the rock density (unit: kg/m$^3$), $D$ is the detonation velocity (unit: m/s), and $r_c$ is the explosive roll radius (unit: m).

In Equation (3), $\alpha$ is the stress wave attenuation index.

$$\alpha = 2 - \mu/(1-\mu) \tag{6}$$

where $\mu$ is the Poisson's ratio of the rock.

The blast hole is close to the existing tunnel lining in this project. The calculation formula of the fracture zone in the surrounding rock caused by blasting with the relevant blasting parameters is shown in Table 6, and it can be used to determine the fracture zone produced by blasting in the tunnel lining.

**Table 6.** Blasting fracture zone calculation parameters.

| Cement Density $\rho_s$ (kg/m$^3$) | Detonation Velocity $D$ (m/s) | Explosive Roll Radius $r_c$ (mm) | Poisson's Ratio $\mu$ | Lateral Stress Coefficient $\psi$ | Tensile Strength $\sigma_t$ (MPa) |
|---|---|---|---|---|---|
| 2400 | 5000 | 18 | 0.25 | 0.33 | 2.01 |

The parameters are substituted into Equations (3)–(6), and the theoretical radius of the blasting fracture zone in the tunnel lining is calculated to be 0.86 m, which is greater than 0.7 m (the thickness of the lining structure). The results show that the blasting load leads to penetrating cracks in the tunnel lining, which are beneficial for dividing the tunnel

lining into small pieces. Moreover, the seam pre-cut in the lining by the machine greatly weakens the resistance of the tunnel lining. The subway tunnel lining will collapse under the action of the blasting impact stress and the gravity of the surrounding rock, which makes it possible to quickly remove the tunnel lining.

### 4.2.3. Stress Analysis of Surrounding Rock

The blasting stress evolution laws are analyzed with time after the dynamite blast, and the time history curves of the absolute value of the maximum principal stress of the surrounding rock at different times in the blasting process are shown in Figure 9.

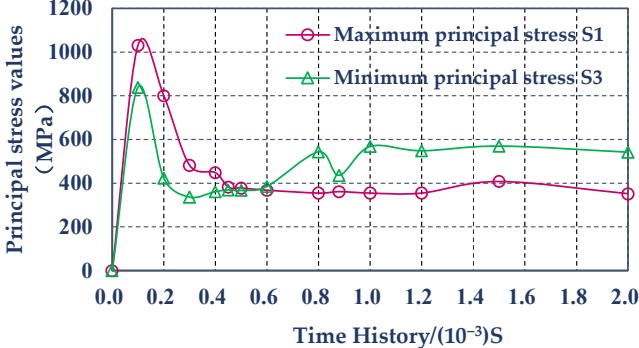

**Figure 9.** The absolute value of principal stress evolution law after blasting.

From the initiation of the explosive to the time of $0.1 \times 10^{-3}$ s, the maximum principal stress $S_1$ and the minimum principal stress $S_3$ gradually increase, which is caused by the explosion. The $S_1$ stress induced by blasting is 1030 MPa and reaches the maximum value at the time of $0.1 \times 10^{-3}$ s, and the $S_3$ stress is 886 MPa. The surrounding rock is sandy mudstone with a compressive strength of 24.7 MPa, and it is damaged by the action of the blasting stress based on the maximum tension theory. The $S_1$ stress and the $S_3$ stress gradually decrease from the time of $0.1 \times 10^{-3}$ s to $0.4 \times 10^{-3}$ s. The $S_1$ stress tends to be stable, but the $S_3$ stress increases and the value is greater than that of the $S_1$ stress until the time of $0.6 \times 10^{-3}$ s. Lastly, the $S_3$ stress tends to be stable at the time of $1.0 \times 10^{-3}$ s. The above blasting stress evolution law shows that the explosion pressure is released rapidly after the dynamite blast, and the $S_1$ stress decreases to a stable level after reaching the maximum value. However, the $S_3$ stress decreases after reaching the maximum value and then increases to a stable value by the superposition of the blasting stress.

### 4.2.4. Vibration Velocity Analysis

According to the propagation law of blasting waves [26–28], set up five vertical monitoring points that were taken every 10 m downward from the ground surface on the tunnel vault (the center of the tunnel cross-section); five horizontal monitoring points are taken every 10 m on the ground surface from the right side of the tunnel axis, as shown in Figure 4. The peak particle velocity of the measuring points caused by the blasting load at different vertical depths and horizontal distances are listed in Table 7.

It can be seen in Table 6 that the point with the maximum PPV is closest to the explosion center in the vertical direction, and the PPVs in the directions of X, Y, and Z are 7.83 cm/s, 7.36 cm/s, and 4.82 cm/s, respectively, with a vector combined vibration velocity of 11.77 cm/s. The PPVs gradually decrease as the distance from the explosion center is in the vertical direction. However, the PPV of the vertical point $V_1$ (on the ground surface) increases again because of the influence of the freedom degrees of the ground surface, which are one more than that of the surrounding rock. For the horizontal measuring point, the PPVs in the directions of X, Y, and Z are 3.0 cm/s, 4.15 cm/s, and 4.17 cm/s, respectively, with a vector combined vibration velocity of 5.33 cm/s. The PPVs of the horizontal measuring points on the surface decrease with the increase in the distance between the tunnel vaults. The measuring point $H_5$ is 40m away from the explosion central axis, the PPVs

decrease to 2.4 cm/s. Therefore, the PPVs of the adjacent non-expanded tunnel structure within 80 m of the explosion center do not exceed the control value of 5.0 cm/s.

**Table 7.** The PPVs of measuring points.

| Measuring Points | | Explosion Center Distance (m) | $V_x$ (cm/s) | $V_y$ (cm/s) | $V_z$ (cm/s) | $V_r$ (cm/s) |
|---|---|---|---|---|---|---|
| Vertical measuring points | $V_1$ | 47.2 | 3.00 | 4.15 | 4.17 | 5.33 |
| | $V_2$ | 37.2 | 2.24 | 2.12 | 3.02 | 3.45 |
| | $V_3$ | 27.2 | 2.45 | 3.04 | 3.6 | 4.36 |
| | $V_4$ | 17.2 | 5.44 | 4.06 | 3.08 | 6.71 |
| | $V_5$ | 7.2 | 4.82 | 7.83 | 7.36 | 11.77 |
| Horizontal measuring points | $H_1$ | 0 | 3.00 | 4.15 | 4.17 | 5.33 |
| | $H_2$ | 10 | 2.24 | 1.05 | 2.70 | 3.53 |
| | $H_3$ | 20 | 1.97 | 2.12 | 2.02 | 2.92 |
| | $H_4$ | 30 | 1.46 | 1.97 | 2.47 | 2.83 |
| | $H_5$ | 40 | 0.72 | 1.03 | 2.13 | 2.40 |

The relationship between the PPVs and the distance from the blasting center is obtained by the fitting of the calculation data (the PPVs of the surface measuring point are not used considering the effect caused by the boundary freedom degrees) is shown in Figure 10.

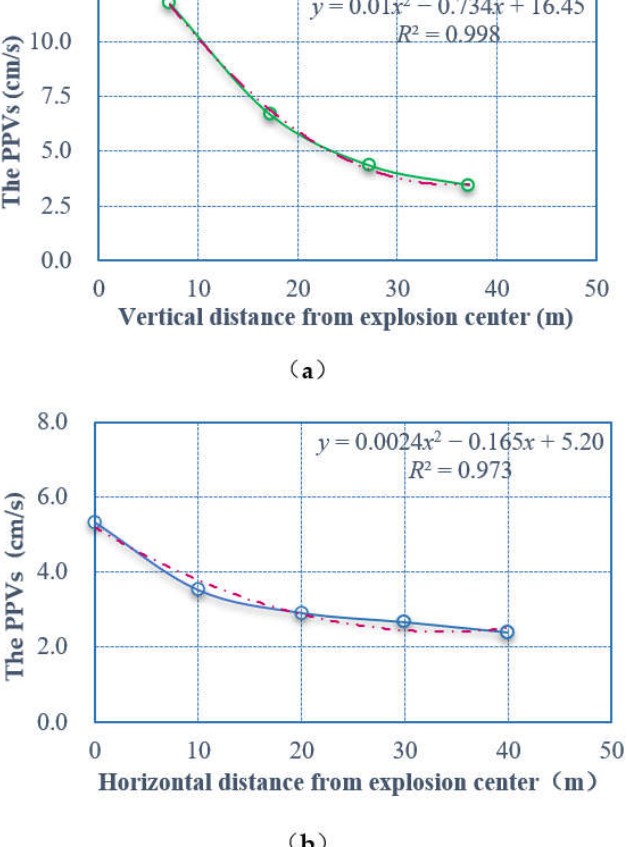

（**a**）

（**b**）

**Figure 10.** The relationship between PPVs and distance from blasting center: (**a**) the PPV evolution law at different vertical depths; (**b**) the PPV evolution law at different horizontal distances on the surface.

The relationships between the PPVs and the vertical distance H and the horizontal distance L from the explosion center are as follows:

$$V_H = 0.01H^2 - 0.734H + 16.45 \tag{7}$$

$$V_L = 0.0024L^2 - 0.165L + 5.20 \tag{8}$$

Under the condition of the same stratum and charge quantity, the PPVs at different positions from the blasting center can be predicted using Equations (7) and (8) so as to provide a theoretical and design basis for the in situ expansion of subway tunnels by blasting.

## 5. Conclusions and Discussion

Based on a practical engineering case of the in situ expansion excavation of an existing subway tunnel to a subway station, the CD construction process of the drilling and blasting method is studied using a numerical simulation, and a rapid demolition method of the existing tunnel lining is proposed. The rationality of the blasting parameters used in this paper is verified from three angles, namely, the overbreak and underbreak of the surrounding rock, the PPVs of the surrounding rock, and the mechanical response of the existing tunnel lining during the demolition process. The following conclusions are drawn:

1.  Using the blasting parameters proposed in this paper to in situ expansion excavate an existing subway tunnel to a subway station, the maximum overbreak of the tunnel vault, side wall, and inverted arch are 15 cm, 7 cm, and 5 cm, respectively, meeting the requirements of tunnel overbreak and underbreak quality.
2.  The blasting tensile stress is greater than the tensile strength of the concrete material, and the radius of the blasting fracture zone is 0.93 m, which is greater than the thickness of the tunnel lining, so penetrating cracks occur in the tunnel lining. Therefore, the existing subway tunnel lining will collapse under the action of the blasting wave load and the gravity of the surrounding rock, which makes it possible to quickly demolish the tunnel lining.
3.  The maximum principal stress $S_1$ and the minimum principal stress $S_3$ produced by the blasting reach the limit values of the surrounding rock, with the $S_1$ stress being 1030 MPa and the $S_3$ stress being 886 MPa from the initiation of the explosion to the time of $0.1 \times 10^{-3}$ s. The surrounding rock is damaged by the action of the blasting stress. The $S_1$ stress and the $S_3$ stress gradually decrease from the time of $0.1 \times 10^{-3}$ s to $0.4 \times 10^{-3}$ s. The $S_1$ stress tends to be stable; the $S_3$ stress increases and the value is greater than that of $S_1$ until the time of $0.5 \times 10^{-3}$ s. Lastly, the $S_3$ stress tends to be stable at the time of $0.8 \times 10^{-3}$ s.
4.  The PPVs decrease with the increase in the distance from the blasting center. The relationships between the PPVs and the vertical distance H and the horizontal distance L of the explosion center are as follows: $V_H = 0.01H^2 - 0.734H + 16.45$ and $V_L = 0.0024L^2 - 0.165L + 5.2$. Under the condition of the same stratum and charge quantity, the PPVs at different positions from the blasting center can be predicted using Equations (4) and (5) so as to provide a theoretical and design basis for the in situ expansion of tunnel blasting.

**Author Contributions:** Conceptualization, J.Z. and B.Z.; investigation, P.S. and B.D.; methodology, B.Z., P.S. and Y.W.; resources, P.S.; data curation, J.Z. and P.S.; writing, J.Z. and B.Z.; formal analysis, B.D. and Y.W.; writing—review and editing, J.Z., B.D. and Y.W.; funding acquisition, B.Z.; project administration, B.Z. All authors have read and agreed to the published version of the manuscript.

**Funding:** This work was supported by the Chongqing Doctor Direct Online Fund (Nos. CSTB2022BSXM-JSX0024), the Chongqing Education Commission Science and Technology Fund (Nos. KJQN202101506), and the State Key Laboratory of Mountain Bridge and Tunnel Engineering Open Foundation (Nos. SKLBT-2110).

**Institutional Review Board Statement:** Not applicable.

**Informed Consent Statement:** Not applicable.

**Data Availability Statement:** The data used to support the findings of this study are available from the corresponding author upon request.

**Conflicts of Interest:** The authors declare no conflict of interest.

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
