# Peer review of "A Finite Element Analysis of Tunnel Lining Demolition by Blasting for Subway Tunnel Expansion"

_applsci, doi:10.3390/app12199564_

Round 1

Reviewer 1 Report

This paper aims to present a method for expanding subway tunnels in case new modifications and platforms are required. The topic is interesting and the authors showed a great effort to present the required analysis for a successful demolition of the studied case.

However, the manuscript should be completely revised:

The English editing is required. There are frequent grammatical mistakes and the style is not appropriate (e.g. what is more should be replaced by moreover or furthermore).

A material and method section is required to completely explain both experimental and numerical methods. It should be explained how different parameters including compressive strength of the sandstone, K or alpha in equation 1 are measured.

In lines 138 to 140 it is stated that "Based on numerical simulation calculation, the optimized blasting parameters are determined as follows:". Which numerical modelling is used here? it should be completely explained in the added material and methods section.

Lines 225 to 226 are giving information about mechanical properties of the cement, this information should be provided before lines 191 to 193.

In figure 7 a colour coded legend is required for showing the amount of principal stresses in MPa. The information in the text cannot be followed  in this figure.

The presented values in Table 5 should be double checked. The compressive stress values seem to be very low.

Numbering of the equations 3 and 4 should be double checked.

It is not clear that the referenced values in Table 6 are related to the sandstone or the cement. This should be organised in the added material and methods section as well.

The references should be checked again. At least a reference is listed twice. Some references are different in the text and reference list (e.g. Zhang et al. [19-23]).

Author Response

  • is not appropriate (e.g. what is more should be replaced by moreover or furthermore).

Response: Thank you for your suggestion. We Have revised in the manuscript.

  • A material and method section is required to completely explain both experimental and numerical methods. It should be explained how different parameters including compressive strength of the sandstone, K or alpha in equation 1 are measured.

Response: The construction site does not have the condition of testing the K or alpha values. So We selected the K or alpha values by the similar surrounding rock in the manuscript.

  • In lines 138 to 140 it is stated that "Based on numerical simulation calculation, the optimized blasting parameters are determined as follows:". Which numerical modeling is used here? it should be completely explained in the added material and methods section.

Response: Has been revised in the manuscript.

  • Lines 225 to 226 are giving information about the mechanical properties of the cement, this information should be provided before lines 191 to 193.

Response: Has been revised in the manuscript

  • In figure 7 a color-coded legend is required for showing the number of principal stresses in MPa. The information in the text cannot be followed in this figure.

Response: For a more convenient dimensional analysis and calculation, we established the model in the software selected cm-g-ms. So the pressure unit needs to be converted In figure 7.

  • The presented values in Table 5 should be double-checked. The compressive stress values seem to be very low.

Response: The blasting footage of the existing subway tunnel in expanding expansion construction is 1.5m, and the charge quantity of per hole is 1 kg. The compressive stress affected by the blasting stress decays with time and the interface reflection between the tunnel lining and surrounding rock.

  • Numbering of the equations 3 and 4 should be double-checked.

Response: The equations 3,4, and 5 has been checked and revised in the manuscript.

  • It is not clear whether the referenced values in Table 6 are related to the sandstone or the cement. This should be organized in the added material and methods section as well.

Response: Has been revised in the manuscript.

  • The references should be checked again. At least a reference is listed twice. Some references are different in the text and reference list (e.g. Zhang et al. [19-23]).

Response: Has been revised references in the manuscript.

Reviewer 2 Report

 the paper is worthy and good. The following comments are suggested:

1- the title of the paper mey be revised as " A finite element analysis of Tunnel Lining Demolition by Blasting for a Subway Tunnel  Expanding" 

2- the sentences in the abstract are too lengthy and confusing. please revise the English of the abstract.

3- its better not to use the abbreviates of the phrases when appeqaring for the first time in the abstract. for example CD excavation method.

4- Please change the "particle vibration velocity (PPV)" to "particle peak velocity (PPV)' in the abstract, keywords and text.

5- the literature review on tunnel construction and blasting modeling may be improved. especially various numerical methods such as finite difference and boundary elements may be used, for example check the following papers:

V. Sarfarazi, Hadi Haei, Salman Safav, M F Marji, Zheming Zhu, 2019, Interaction between two neighboring tunnel using PFC2D, Structural Engineering and Mechanics 71 (1), 77-87.

M. S. Abdollahi, M. Najafi, AR Yarahmadi Bafghi, MF Marji, 2019,  A 3D numerical model to determine suitable reinforcement strategies for passing TBM through a fault zone, a case study: Safaroud water transmission tunnel, Iran, Tunneling and Underground Space Technology 88, 186-199

M Lak, MF Marji, AY Bafghi, A Abdollahipour, A Coupled Finite Difference-Boundary Element Method for modeling the propagation of explosion-induced radial cracks around a well bore, Journal of Natural Gas Science and Engineering 64 (1), 41-51.

6- Please replace "detonation velocity (D)" for "explosion speed (D)" in the text.

7- please show the Poisson's ratio as given in Table 3 for the equation (5) and Table 6. There should be only one alphabet for showing Poisson's ratio in your text. Please check for the rest symbols too. 

8- the first two sentences in the conclusion are also lengthy and confusing please rewrite them.

Author Response

  • The English editing is required. There are frequent grammatical mistakes and the style is not appropriate (e.g. what is more should be replaced by moreover or furthermore).

Response: Thank you for your suggestion. We Have revised in the manuscript.

  • A material and method section is required to completely explain both experimental and numerical methods. It should be explained how different parameters including compressive strength of the sandstone, K or alpha in equation 1 are measured.

Response: The construction site does not have the condition of testing the K or alpha values. So We selected the K or alpha values by the similar surrounding rock in the manuscript.

  • In lines 138 to 140 it is stated that "Based on numerical simulation calculation, the optimized blasting parameters are determined as follows:". Which numerical modeling is used here? it should be completely explained in the added material and the title of the paper may be revised as " A finite element analysis of Tunnel Lining Demolition by Blasting for a Subway Tunnel Expanding"

Response: The title of the paper has been revised

  • the sentences in the abstract are too lengthy and confusing. please revise the English of the abstract.

Response: The abstract has been revised in the manuscript.

  • it is better not to use the abbreviates of the phrases when appearing for the first time in the abstract. for example CD excavation method.

Response: Has been revised in the manuscript

  • Please change the "particle vibration velocity (PPV)" to "particle peak velocity (PPV)' in the abstract, keywords and text.

Response: Has been replaced in the manuscript.

  • the literature review on tunnel construction and blasting modeling may be improved. especially various numerical methods such as finite difference and boundary elements may be used, for example check the following papers:
  1. Sarfarazi, Hadi Haei, Salman Safav, M F Marji, Zheming Zhu, 2019, Interaction between two neighboring tunnel using PFC2D, Structural Engineering and Mechanics 71 (1), 77-87.
  2. S. Abdollahi, M. Najafi, AR Yarahmadi Bafghi, MF Marji, 2019,  A 3D numerical model to determine suitable reinforcement strategies for passing TBM through a fault zone, a case study: Safaroud water transmission tunnel, Iran, Tunneling and Underground Space Technology 88, 186-199

 M Lak, MF Marji, AY Bafghi, A Abdollahipour, A Coupled Finite Difference-Boundary Element Method for modeling the propagation of explosion-induced radial cracks around a wellbore, Journal of Natural Gas Science and Engineering 64 (1), 41-51.

Response: We have added above this paper, and has been revised references in the manuscript.

  • Please replace "detonation velocity (D)" for "explosion speed (D)" in the text.

Response: has been replaced in the text.

  • please show the Poisson's ratio as given in Table 3 for equation (5) and Table 6. There should be only one alphabet for showing Poisson's ratio in your text. Please check for the rest symbols too.

Response: Has been revised in the manuscript.

  • the first two sentences in the conclusion are also lengthy and confusing please rewrite them.

Response: The conclusion has been revised in the manuscript.

Round 2

Reviewer 1 Report

Thanks for reviewing the manuscript.

Reviewer 2 Report

the paper is revised but minor revisions may be required regarding the English of the article for example the first and the last sentences of the abstract should be revised. these sentences are not clear and their  grammar is also mixing up.  

Please kindly improve the English of the first sentence of the first paragraph in conclusion and discussion part of the paper too.